# Optimization of Hydrologic Response Units (HRUs) Using Gridded Meteorological Data and Spatially Varying Parameters

**David Poblete [1],\*** , **Jorge Arevalo [2,3]** , **Orietta Nicolis [4,5] and Felipe Figueroa [6]**

[1]    School of Civil Engineering, Universidad de Valparaíso, Valparaíso 2340000, Chile
[2]    Department of Meteorology, Universidad de Valparaíso, Valparaíso 2340000, Chile; jab@meteo.uv.cl
[3]    Department of Hydrology and Atmospheric Sciences, University of Arizona, Tucson, AZ 85721, USA
[4]    Faculty of Engineering, Univesidad Andres Bello, Viña del Mar 2520000, Chile; orietta.nicolis@unab.cl
[5]    Research Center for Integrated Disaster Risk Management (CIGIDEN), ANID/FONDAP/15110017, Santiago 8320000, Chile
[6]    Plataforma de Investigación en Ecohidrología y Ecohidráulica (EcoHyd), Santiago 8320000, Chile; felipe.figueroaba@gmail.com
**\***    Correspondence: david.poblete@uv.cl

**Abstract:** Although complex hydrological models with detailed physics are becoming more common, lumped and semi-distributed models are still used for many applications and offer some advantages, such as reduced computational cost. Most of these semi-distributed models use the concept of the hydrological response unit or HRU. In the original conception, HRUs are defined as homogeneous structured elements with similar climate, land use, soil and/or pedotransfer properties, and hence a homogeneous hydrological response under equivalent meteorological forcing. This work presents a quantitative methodology, called hereafter the principal component analysis and hierarchical cluster analysis or PCA/HCPC method, to construct HRUs using gridded meteorological data and hydrological parameters. The PCA/HCPC method is tested using the water evaluation and planning system (WEAP) model for the Alicahue River Basin, a small and semi-arid catchment of the Andes, in Central Chile. The results show that with four HRUs, it is possible to reduce the relative within variance of the catchment up to about 10%, an indicator of the homogeneity of the HRUs. The evaluation of the simulations shows a good agreement with streamflow observations in the outlet of the catchment with an Nash–Sutcliffe efficiency (NSE) value of 0.79 and also shows the presence of small hydrological extreme areas that generally are neglected due to their relative size.

**Keywords:** hydrologic response units; principal component analysis; hierarchical cluster analysis; PCA/HCPC method

## 1. Introduction

The concept of Hydrologic Response Units (HRU) has risen as one of the most common approaches for semi-distributed hydrological modelling [1,2]. Flügel [2] defined an HRU as a homogeneous structured element having similar climate, land-use, soil and/or pedotransfer properties, hence a homogeneous hydrological response under equivalent meteorological forcing. An important assumption is that the variation of the hydrological process dynamics within a single HRU is small compared with the hydrologic dynamics and responses to other units defined in the model. Many authors assume that HRU do not necessarily represent contiguous geographical areas so the topology of the elements is simplified or just neglected and the total discharge of the watershed

is calculated as the incremental input of every independent element and propagated to its outlet; assumption that we will also consider in the rest of this study [3,4].

Traditionally, land use/land cover, topographic characteristics and soil types have been used as proxies of many of the parameters involved in the governing equations and parameterizations of the lumped, semi-distributed and even distributed models, but always with a certain degree of uncertainties [5–7]. Most of the methodologies to delineate HRUs rest on the expected relationships between physical-ecological characteristics of the catchment and the corresponding hydrological properties reflected on the hydrological model parameters. Hence, HRUs are usually defined by the superposition of land use and soil type and after the classification, quantitative or qualitative relations are used to estimate hydrologic parameters on each HRU. One of the most common approaches has been to include the sub-basins in the process, hence the intersection of the sub-basins, land use categories and soil type polygons in a GIS represents the minor elements for hydrologic modelling [4,8].

A different approach is used in Savvidou et al. [4], as they estimate the curve number (CN) parameter for reference conditions using soil permeability, vegetation classes and drainage capacity maps and then the HRUs are defined based on the separation of areas according to the CN values. According to the authors, these delineated HRUs can be used in any hydrological model as the Soil Conservation Service SCS-CN model, which is widely used and understood.

Even though it is important to properly define the HRU for a good representation of the hydrological processes and dynamics, methods and tools for identifying an appropriate scale are often missing. The challenge is to identify a proper method for the discretization of the basins, losing the least information possible and maximizing the model reliability and utility that in turn play a crucial role in the accuracy of the models [9–11]. If over simplification of the basin characteristics is done, small areas of extreme hydrologic behavior can be neglected by a lack of representation in the aggregation procedures [11]. On the other hand, if the used data is highly detailed and fragmented, it can lead to an excessive number of HRUs, making the modelling impracticable.

Although meteorological variables are inputs to every model, none of the methodologies use that information directly in the construction process of the HRU. Flügel [2] suggested more than two decades ago that the use of meteorological information to construct HRU is advisable, but it has not been explored in depth probably due to the lack of good quality spatial meteorological information. Today, this idea is more plausible and can be considered because one of the basic assumptions on HRU is that meteorological forcing is homogeneously spatialized over the domain of the HRU. Therefore, the spatial heterogeneity of the precipitation and other variables can be incorporated in the delineation of HRUs. An indirect approach to include climate information is used by Young et al. [12], where 15 watersheds of the Sierra Nevada in California are discretized in HRU by the intersection of sub-basins, soils type, vegetation cover and elevation bands in the Water Evaluation And Planning System model (WEAP; [13]). They calculate fractional areas for each sub-basin using a vegetation cover/soil type combination in 250 m elevation bands ranging from 500 to 4000 m above sea level, in order to provide a finer discretization for snow accumulation and melt modelling. This has been a common practice in the use of this model in semi-arid basins in Chile (for instance, [14,15]).

Given all these issues, some questions arise: How to use the detailed information available on land use, geomorphologic properties and climatic behavior for the separation of a manageable number of independent modelling units? Which criteria must be used to simplify the complexity of hydrologic dynamics of a watershed into the smallest number of homogeneous units as possible without losing valuable information? Does the use of these independent modeling units ensure heterogeneity of hydrologic response between them?

This paper presents a quantitative methodology for the determination of unstructured HRUs based on the homogeneity of the hydrological parameters used by any specific hydrological model and its meteorological inputs. The method was named the principal component analysis and hierarchical cluster analysis (PCA/HCPC) method, as principal component analysis (PCA) is performed in order to get an independent set of vectors, the principal components, to be used in a hierarchical cluster (HC)

algorithm to obtain the desired independent HRUs. The result minimizes the internal variability of hydrologic properties in each HRU and simultaneously maximizes the variability between different HRUs, and subsequently of the hydrologic responses of each element.

To test the PCA/HCPC Method, HRU delineation is performed for the Alicahue river basin, an Andean semi-arid basin located in Central Chile. Hydrologic parameters and climate averaged values used by the semi-distributed WEAP model (Water Evaluation and Planning System) are calculated over a regular grid, that in turn are used to classify each cell in the mentioned HRUs. Climate variables are based on a 1km resolution bias-corrected model output for three periods of 12-month using the WRF model [16] and the hydrologic parameters are estimated by topographic characteristics derived from 30m ASTER DEM [17] and Land Use data from Natural Resources Research Center of Chile [18]. Finally, the performance, accuracy and skill of the model using ten different configurations of HRU are analyzed using common modelling indicators.

## 2. Methodology

The PCA/HCPC method consists in the creation of a dataset of raster files comprising hydrologic parameters and meteorological variables used by the target hydrological model. Then, through principal component and hierarchical cluster analyses, every cell of the raster files is classified into a specific cluster to form the different HRUs.

### 2.1. Area of the Study Case

Central Chile has a landscape with a very complex topography. It is surrounded by the high peaks of the Andes Mountains, usually above 4000 m.a.s.l., in the east and the Pacific Ocean only about 150 km west of the mountains. Most of the river basins in this area have a latitudinal preferential path, downstream of the Andes up to the ocean. Its climate corresponds to the Mediterranean, with dry summers, and temperatures are usually mild, ranging from about 0 °C as a minimum during winter up to 35 °C as a maximum during summer, except for the high elevation lands where below freezing temperatures are usual during winter. Mean annual precipitation is about 400 mm for the valleys and coastal areas, which is mostly due to winter frontal storms, hence the spatial variability is mostly modulated by the orographic effects.

The domain of the area of study corresponds to the Alicahue River Basin, which is located between geographical coordinates 32.39° S to 32.21° S in latitude and 70.76° E to 70.41° E in longitude, in the province of Petorca, Valparaiso Region, Central Chile (Figure 1). This is a sub-catchment of the La Ligua River Basin, that receives water from other minor streams and flows into the sea, with a total length of nearly 200 km. The Alicahue River has a length of just 30 km and its drainage area is just 354 km$^2$, but its topography ranges from 780 m.a.s.l. up to 3985 m.a.s.l., with almost half of its area located above 2500 m.a.s.l. In winter and during rainfall, the 0 °C isotherm in Central Chile is typically located at about 2500 m.a.s.l. [19], allowing snow accumulation in most parts of the Andes Mountains. Hence, at the outlet of the basin, there is an important flow between mid-spring and the beginning of summer in the southern hemisphere (from October to January) due to snow melting. Agriculture uses the waters from La Ligua River, but most of the discharge of this river during the dry season comes from upper basins, such as the Alicahue River, where snow accumulation is possible during the cold and wet winter.

The Alicahue River Basin is a small catchment and has limited human intervention, which simplifies the analysis of the results. Additionally, its behavior is comparable to several other high-altitude catchments in Central Chile, where complex topography is dominated by the traversal valleys downstream of the Andes Mountains and snowmelt is one of the dominant hydrologic drivers.

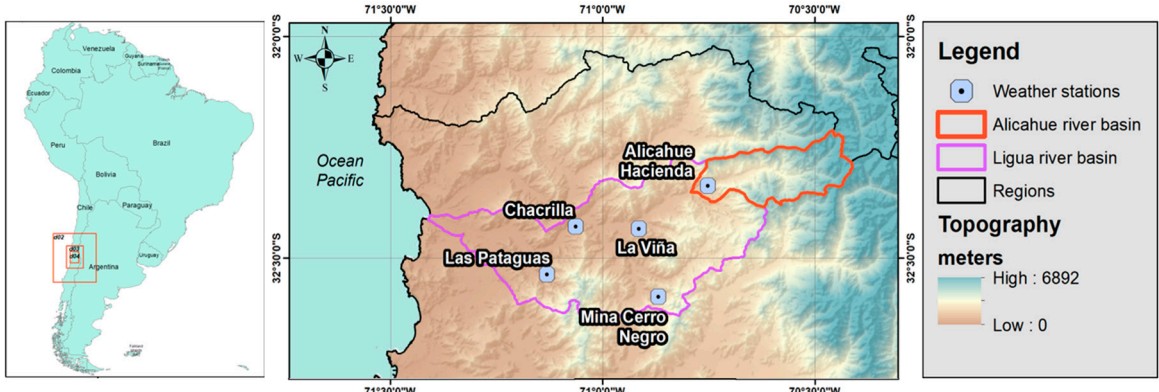

**Figure 1.** Area of study. Relative location in South America and Weather Research and Forecasting (WRF) domains (left), topography of the area of study indicating the limits of Alicahue and La Ligua River Basins (red and purple polygons, respectively) and the location of the weather stations (streamflow station co-located with Alicahue Hacienda weather station).

The only available stream gauge station is located in the outlet of the Alicahue River Basin (32.20° S, 70.45° W), from station BN 05200001-7 "Rio Alicahue en Colliguay", from the General Directorate of Water (DGA in Spanish), with a recording period starting in 1963.

Although only the station Alicahue Hacienda is located inside the basin, the frontal nature of the precipitation makes it reasonable to correlate near observations. The values recorded at these stations are 1.16 m$^3$/s for mean annual streamflow, 267 mm for total annual precipitation and 15.1 °C for mean annual temperature.

## 2.2. Hydrologic Parameters and Meteorological Datasets

The PCA/HCPC method uses raster maps of the hydrologic parameters and mean annual values of the meteorological variables used by the chosen hydrologic model. These maps need to be constructed or generated previously by any methodology.

In the case of this study, the WEAP model [13] is used to test the methodology. WEAP is a water allocation model that has been used for water resources management in several studies over several catchments around the world (for instance [12,20]) and particularly in Chile ([14,15]). It has incorporated a hydrology module that represents the mass balance in elements, called catchments, in which simplified hydrological fluxes and storages are modeled using a one-dimensional and two-layer storage system. Although a WEAP catchment can be used as a single HRU, the catchment element can be internally divided in more separate units, each of them as a single HRU. The methodology most widely used divides the catchment into elements by land use cover within a given elevation band and sub-basin, with all the HRUs having the same meteorological condition on each elevation band.

The upper layer of the catchment element has four hydrologic parameters: Soil water capacity (Sw, in mm) represents the soil layer depth; runoff resistance factor (RRF) is equivalent to the run-off coefficient in the rational equation; root zone conductivity (Ks, in mm/month) corresponds to the saturated hydraulic conductivity in the soil layer; and kc to the crop coefficient of the vegetation. The parameter preferred flow direction (f, no units) controls the water flowing from the upper layer to the lower layer as interflow or deep percolation (f = 1 for total horizontal flow and f = 0 for total vertical flow). Dw and kd represent the depth and the saturated hydraulic conductivity of the deeper layer of the catchment element, respectively. Finally, the simple snow model uses two temperature thresholds, for melting and freezing (Tl and Ts, respectively), totalizing nine parameters (for detailed information on the water balance equations, see [13]).

Figure 2 shows the maps of the WEAP parameters f, RRF and the log values of Sw and Ks, respectively, for the Alicahue River Basin. RRF (top left) is related to milder slopes and vegetated terrain; f (top right) depends on the terrain slope and soil properties. Sw (bottom left) and Ks (bottom

right) are shown on a log scale for better visualization. Sw presents the deepest soils in the flat lands near the main water course, in contrast to the sides of the hillslope. The latter is also related to the vegetation land cover and slope and dominated by very permeable areas in high-altitude wetlands.

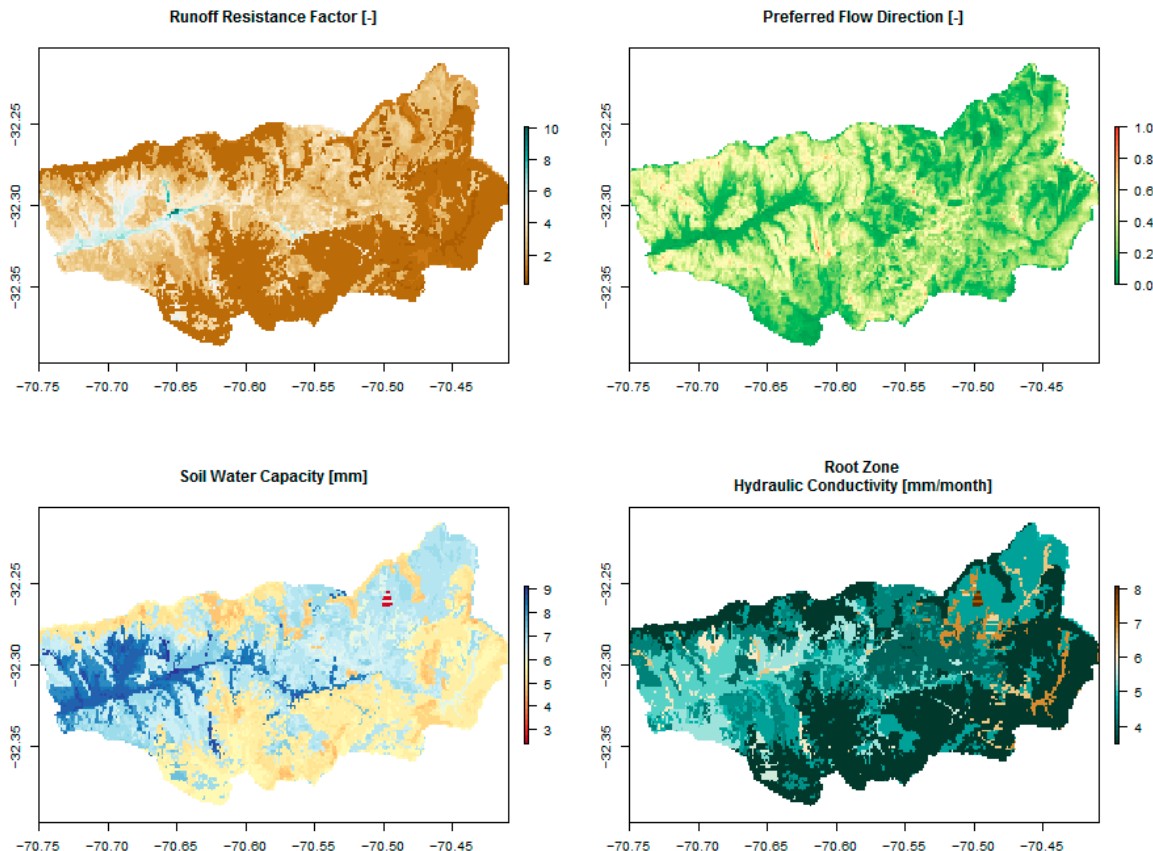

**Figure 2.** Parameter map covering the Alicahue River Basin for: (**top left**) Runoff Resistance Factor (RRF), (**top right**) Preferred Flow Direction (f), (**bottom left**) Soil Water Capacity (Sw) and (**bottom right**) Root Zone Hydraulic Conductivity (Ks). Sw and Ks are plotted on a log scale.

As the spatial representativeness of meteorological stations is small in complex terrain and observations are usually scarce, the Weather Research and Forecasting (WRF) model version 3.4.1 was used to simulate three periods of 12 consecutive months each: (1) 1 July 2003 to 30 June 2004, (2) 1 June 2009 to 31 May 2010 and (3) 1 June 2010 to 31 May 2011. Those simulations were performed in four nested domains with 27, 9, 3 and 1 km of horizontal resolution and 50 terrain-following vertical levels, the innermost domain (250 × 202 grid points at 1 km resolution) was used for the analysis and entirely covered the study area, as can be seen in Figure 1.

The WRF model is used to perform a dynamical downscaling of larger-scale models. For this purpose, the Advanced Research WRF (ARW) dynamical core was selected to solve the grid-scale thermodynamic equations governing the atmosphere, while the sub-grid processes were parameterized using the WRF single moment scheme for the cloud microphysics, the Rapid Radiative Transfer Model for Global (RRTMG) scheme for both shortwave and longwave radiation, the Noah scheme as a land-surface model, the quasi-normal scale elimination scheme for the boundary and surface layer processes and the Kain–Fritsch scheme (only in the two outer domains) for the convection parameterization.

Figure 3 shows the mean annual precipitation and the mean temperature for the 36 months of WRF simulations. Other climatological variables, such as relative humidity, net radiation, albedo, evapotranspiration and wind speed, are not shown.

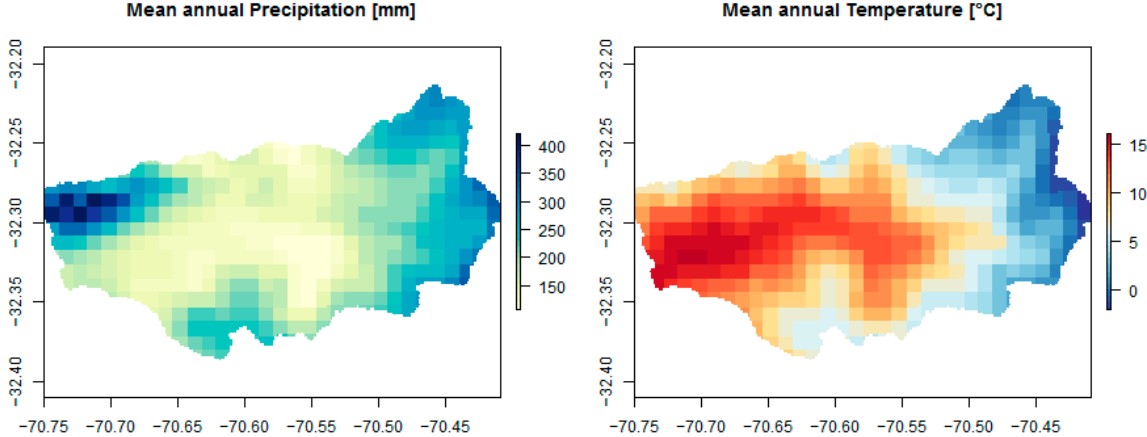

**Figure 3.** Maps covering the Alicahue River Basin for: (**left**) Mean annual precipitation, (**right**) mean temperature.

As the WRF simulation period covered only 36 non-continuous months, the variables from that simulation were not suitable to drive the long-term hydrological modeling. Hence, a simple relationship between the available observed precipitation time series (from weather stations near the basin, see Figure 1) and the modeled areal average in each HRU was used to force the longer-term WEAP model runs. Long-term temperature time series were extrapolated to the HRUs using a simple linear model between the mean annual temperature modeled in WRF and "Alicahue Hacienda" station records using the variables elevation, aspect, mean longitude and mean latitude for the HRUs in each of the simulations.

As both types of datasets (i.e., parameters and meteorology) have different cell sizes and extension, to join both datasets, the meteorological raster maps are resampled from a common grid system into the parameter base grid by the nearest neighbor method using the Vincenty (ellipsoid) great circle distance from the distm function of the geosphere package in R [21].

### 2.3. Clustering Processes and HRU Delineation

In this section, the core of the HRU delineation process is detailed. The gridded model-specific parameters and the climatological information are used in the principal component analysis to then use its first components in the hierarchical clustering.

The principal component analysis (PCA) technique consists of describing a multidimensional dataset of $p$ variables and $n$ individuals, using a smaller number of orthogonal vectors (the principal components) that incorporate as much information from the of the original dataset as possible. Denoting by $X$ the $n \times p$ original matrix, composed by p-n-dimensional vectors, the PCA seeks the linear combinations of the columns of $X$ with maximum variance. Such linear combinations are given by $Xa$ where $a_1, a_2, \ldots a_p$ are constants, and the variance of $Xa$ is $a'Sa$, where $S$ is the sample covariance matrix and ' denotes transpose. The maximization of variance is solved by using the Lagrange multiplier $\lambda$. Simple mathematical calculations prove that **a** must be a (unit-norm) eigenvector, and $\lambda$ the corresponding eigenvalue of the sample covariance matrix $S$ [22]). The $p \times p$ covariance matrix $S$ has exactly $p$ real eigenvalues $\lambda_1, \lambda_2, \lambda_p$, and their corresponding eigenvectors can be defined to form an orthonormal set of vectors. Using the Lagrange multiplier approach, the full set of eigenvectors of $S$ are the solutions to the problem of obtaining up to $p$ new linear combinations (principal components) which successively maximize variance, subject to uncorrelatedness with previous linear combinations [23]. Uncorrelatedness results from the fact that the covariance between two linear combinations, $Xa_k$ and $Xa_{k'}$, is zero for $k \neq k'$ given the orthogonality of the eigenvectors $a_k$ and $ak'$ of the matrix $S$.

Since the variance depends on units of measurement of the variables and the principal componenets in the covariance matrix, it is common practice to standardize the variables before the use of PCA,

that is, each variable is centered and divided by the standard deviation. The PCA technique is widely used for the reduction of dimensionality in environmental datasets (see, for example [24] and the references within). Applications of the PCA to meteorological measurements is described in the works of Joliffe [23,25].

The Principal Component Analysis is performed using the function PCA from the FactoMineR package [26] for multivariate data analysis in R [27] by assigning greater weights to the most important variables, in order to capture more variance of these variables. In this study, precipitation and temperature variables were weighted by a factor of two given its importance in the water balance equation, while the rest of the variables had weights equal to one. This allows the use of the expertise of the modeler in assigning more importance to specific variables.

Working with principal components instead on the original data, allows to obtain more stable results in the clustering process. Since the first dimensions (or components) extract the most information from data and the last ones represent the noise [28], the first components accounting for at least 90% of variance are used in the hierarchical cluster analysis function HCPC from the same FactoMineR2 package. The objective is to capture most of the variability of the most important variables and simultaneously not capture the variability of the less important variables or at least represent a minor proportion of the variance of those variables.

The hierarchical clustering used in this work has been implemented using Ward's criterion [28]. Ward's method is based on an agglomerative or "bottom-up" approach, where the clustering starts by considering each observation as a single cluster, and pairs of clusters are merged as one moves up in the hierarchy. The initial cluster distances in Ward's method may be defined by the squared Euclidean distance between the individuals' values and their averages.

By considering a multivariate database composed by *I* spatial individuals (cells) and K variables (both hydrologic parameters and meteorological variables), the total variance of *Q* clusters (with $Q < i$) is evaluated according to its decomposition in the between and within variances, given by:

$$\sum_{k=1}^{K}\sum_{q=1}^{Q}\sum_{i=1}^{N_q}(x_{iqk} - \bar{x}_k)^2 = \sum_{k=1}^{K}\sum_{q=1}^{Q}N_q(\bar{x}_{qk} - \bar{x}_k)^2 + \sum_{k=1}^{K}\sum_{q=1}^{Q}\sum_{i=1}^{N_q}(x_{iqk} - \bar{x}_{qk})^2 \tag{1}$$

where $x_{iqk}$ is the normalized value of the variable k for the individual *i* of the cluster *q*, $\bar{x}_{qk}$ is the mean of the variable k for cluster *q*, $\bar{x}_k$ is the overall mean of variable k (equal to zero if normalized) and $N_q$ is the number of spatial points or cells in cluster *q*. The first member at the right side of the equation represents the between inertia (or between variance) and the second member, the within inertia (within variance).

The importance of this equation is that the total variance of the system remains constant and as the within variance decreases (the clusters become more homogeneous), the between variance increases (the clusters become more and more different from each other).

At each step of the aggregating algorithm, the increase in within variance is minimized (or the increase in the between variance is maximized). This analysis detects groups of individuals with similar characteristics and hydrologic behavior based on parameters and meteorological similitude between cells. Each group of cells belonging to a cluster represents a single HRU to be used in the hydrological model. It is not necessary for cells to be contiguous to belong to the same cluster, as proximity in the space of attributes does not ensure proximity in the geographical space [29], although this may be desirable if contiguous HRUs are to be delineated.

The optimal number of clusters in the data is selected using the clustering tree and is calculated automatically by the function when the within variance reaches a minimum plateau, using the least number of clusters. In the method described by [28], if $\Delta(Q)$ is the between inertia increase when moving from $Q - 1$ to $Q$ clusters, the optimal number of clusters $Q$ is the one which minimizes the relation $\Delta(Q)/\Delta(Q + 1)$. Other indexes to assess the optimal number of clusters are described in [29].

To test the present methodology, we calculate 10 scenarios in which each scenario has a number of s clusters (s from 1 to 10). For each scenario, the method stores for each cell the HRU it belongs to. This is done to evaluate the sensitivity of the hydrological model to the number of HRUs, ranging from a single HRU (a completely lumped model) to a more semi-distributed scheme of the basin with as many HRUs as clusters generated.

*2.4. Hydrological Model Setup and Simulations*

The WEAP model is run using the ten different scenarios described previously. Every configuration of the model uses a different number of HRUs. The lumped configuration was called HRU_01 and uses just one HRU to model the basin. The second scenario uses two HRUs and is called HRU_02 and the rest of the configurations are named similarly depending on the number of HRUs used.

The time series of monthly precipitation and mean monthly temperatures were derived from the meteorological dataset and the observed values recorded in the meteorological stations, as explained in Section 2.2.

The values assigned for the hydrological parameters in each HRU are calculated as the average value for all the cells belonging to the HRUs defined in the previous step. Additionally, the values for wind speed, relative humidity and albedo were set constant for every simulation for simplicity. These values were obtained by intersecting the area for each cluster defined in the previous section with the raster corresponding to the annual mean of each variable obtained from WRF outputs.

The total discharge at the outlet of the catchment is calculated as the aggregation of the discharges produced by each HRU or catchment element in the WEAP modeling in each of the ten scenarios.

For calibration purposes, the WEAP model has spatially constant calibration factors for each of the four parameters assessed in the PCA/HCPC methodology. They are assumed initially as one but can be adjusted in the calibration process to adjust the results of the modeling by the mean of automated or manual techniques.

Finally, the results of the hydrological modeling are analyzed by some common hydrological indicators, such as the Nash–Sutcliffe model efficiency coefficient (NSE) and the root mean square error (RMSE) standardized by the mean discharge.

$$\text{NSE} = 1 - \frac{\sum_{i=1}^{N}(Q_{oi} - Q_{si})^2}{\sum_{i=1}^{N}\left(Q_{oi} - \overline{Q_o}\right)^2} \tag{2}$$

$$\text{RMSE} = \frac{1}{\overline{Q_o}} \cdot \sqrt[2]{\frac{\sum_{i=1}^{N}(Q_{oi} - Q_{si})^2}{N}} \tag{3}$$

where:
$Q_{oi}$ : Observed discharge at time step *i*.
$Q_{si}$ : Simulated discharge at time step *i*.
$\overline{Q_o}$: Average of the observed discharge over the simulation period.

## 3. Results

This section shows the main results in each part of the methodology: (i) The results of the principal component analysis and the hierarchical clustering and (ii) the results of the hydrological modeling using the different schemes of the HRU.

*3.1. PCA and Cluster Analysis*

This section shows the results of the principal component analysis and the hierarchical clustering analysis, the core of the HRU delineation. The PCA was performed over the set of the standardized meteorological variables (precipitation, temperature, relative humidity, wind velocity, albedo and

evapotranspiration) and hydrologic parameters (Sw, f, RRF and Ks). The first result to highlight is that the two first dimensions resulting from the PCA account for 66.8% of the total variance. Adding the following 3rd, 4th and 5th dimensions, they account for 78.7%, 84.6% and 89.6%, respectively, of the total variance of the master dataset. Table 1 shows the variance explained by each consecutive eigenvector or dimension and the contribution of each variable to the dimensions. The first dimension (more than 50% of the total variance) is composed mainly of meteorological variables, with temperature, albedo, wind speed and rainfall being the ones making a greater contribution. The second dimension accounts for more than 16.1% of the total variance and is composed mainly of rainfall and the hydrological parameters.

**Table 1.** Summary from the principal component analysis (PCA) results for the first five dimensions or eigenvectors. The upper part shows the total variance explained by each dimension and the lower, the contribution of each variable to that dimension.

|  | Dim.1 | Dim.2 | Dim.3 | Dim.4 | Dim.5 |
|---|---|---|---|---|---|
| Explained Variance (%) | 50.7 | 16.1 | 11.9 | 5.9 | 5.0 |
| Variables | Contribution to each dimension (%) | | | | |
| Temp | 28.2 | 0.1 | 0.7 | 0.2 | 5.1 |
| Albedo | 11.5 | 1.1 | 5.6 | 1.6 | 10.0 |
| Wind Speed | 10.8 | 1.3 | 0.6 | 0.0 | 8.5 |
| Precipitation | 10.7 | 41.5 | 22.6 | 0.0 | 0.6 |
| Evapotranspiration | 10.2 | 2.2 | 1.5 | 0.5 | 12.3 |
| Net Radiation | 9.7 | 3.5 | 9.6 | 1.4 | 10.7 |
| Relative Humidity | 9.5 | 5.6 | 1.4 | 1.0 | 0.0 |
| Runoff Resistance Coefficient | 4.9 | 13.9 | 10.5 | 12.1 | 2.5 |
| Soil Water Capacity | 4.1 | 20.3 | 6.4 | 4.4 | 9.7 |
| Preferred Flow Direction | 0.3 | 4.7 | 32.1 | 0.4 | 35.6 |
| Root Zone Hydraulic Conductivity | 0.0 | 5.7 | 9.0 | 78.3 | 5.0 |

Table 1 suggests that just the first five dimensions are carrying most of the information, cleaning the statistical noise and, hence, these first five components are used in the HCPC function for the cluster analysis. Based on the new dataset composed of only these principal components, the total variance of the system is fixed for the cluster analysis, therefore, the within variance is expressed as relative to this total hereinafter.

Figure 4 shows the proportion of the within variance relative to the total variance, where the major decrement of the variance occurs up to the case with four clusters and decreases until the case with seven or eight clusters. A decrease in the within variance means that the internal variability of each cluster decreases and, hence, the variance between clusters increases (Equation (1)). As this happens, the hydrologic behavior between HRUs is also expected to be more heterogeneous and simultaneously more homogeneous within each individual HRU, which is expected to lead to a better hydrologic modeling. Hence, it was expected that the optimal number of HRUs for hydrological modeling is four. As described in the methodology, the hierarchical cluster analysis was performed to produce ten scenarios with different numbers of HRUs partitioning the basin, varying between one (lumped model; HRU_1) and ten (HRU_10), to be tested in the hydrological model.

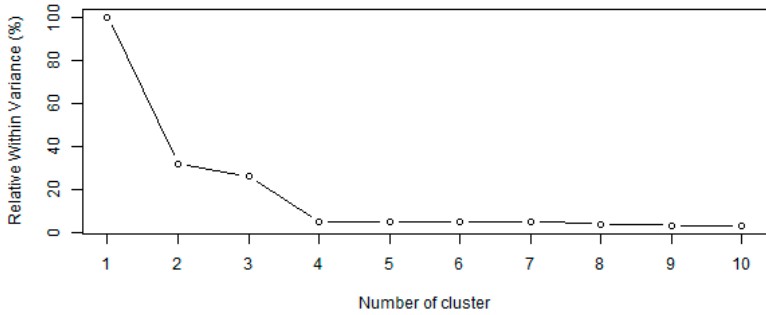

**Figure 4.** Relative within variance for each number of clusters.

Figure 5 shows the distribution of cells in six equal elevation bands, roughly of 550 m each (left plot), as generally used in WEAP as the first step for the separation of HRUs [12,15] and six clusters following the methodology proposed in this work, named PCA/HCPC (right). This number of clusters was chosen because of the best hydrologic results (Section 3.2). Cluster 6 is similar in shape to the highest elevation band as they are concentrated in the eastern part of the basin where the highest elevations are located. For other clusters, the figure shows a clear difference; for instance, cluster 2 in the PCA/HCPC methodology is concentrated in the northwestern part of the catchment, consistent with the high precipitation area identified in Figure 3 (right panel), which is not identified in the traditional elevation bands. The cluster with the lowest elevations (cluster 1) is not as regular as its corresponding elevation band, as this cluster seems to follow the riverbed and the flat riparian zone. Cluster 5 seems to be concentrated in higher and colder areas with Andean vegetation and vegas, characterized by their high water content or retention capacity, compared to the surroundings composed mainly of bare soil and dispersed and small shrubs (cluster 4). Clusters still follow a tendency by elevation, as mean annual temperature is the main variable composing the first dimension, but other variables tend to increase in importance as the number of clusters increases.

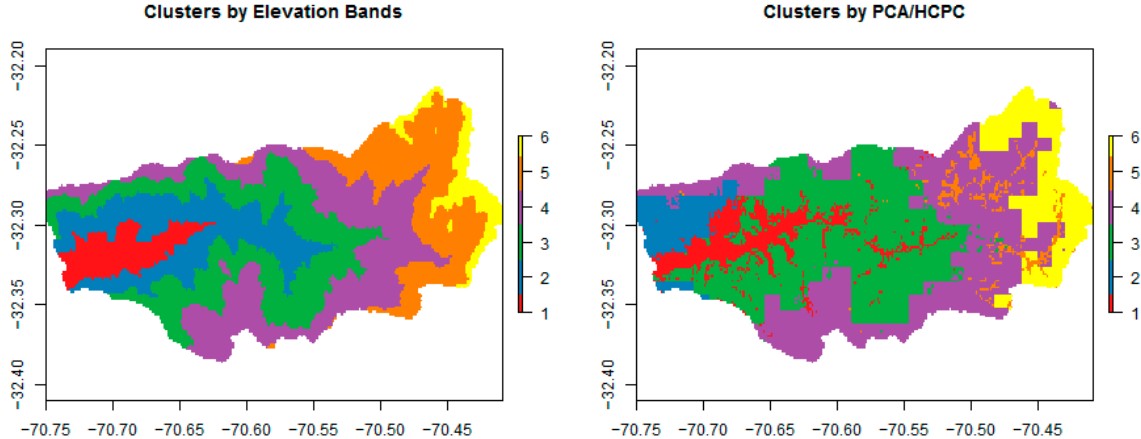

**Figure 5.** Elevations bands every 550 m (**left**), as used in the traditional methodology, and hydrological response unit (HRU) delimitations (**right**) for the simulation with six HRUs by the PCA/HCPC methodology.

Figure 6 shows the boxplot for the values of six selected variables on each cell grouped by cluster to highlight the differences between them. It is possible to observe distinct characteristics between clusters: Cluster 6 is the coldest cluster and has one of the highest precipitation rates; cluster 1 is the warmest, the lowest in altitude, the second in order of dryness and the one with the deepest soil capacity, probably due to its location in the deepest part of the valley, where soils tend to be deeper and have a higher runoff resistance factor, due to the flat terrain and vegetation. Cluster 2 is the one with the highest precipitation. Cluster 5 is the one with the highest hydraulic conductivity, due to the presence of marshes and wetlands. Clusters 3 and 4 are similar, although cluster 4 has a mean value for precipitation of almost 50% more than cluster 3, and also they have differences in the variables not shown. Clusters 6 and 4 have similar hydrologic parameters, but cluster 6 is colder and receives more rainfall.

See online supplemental data available at https://data.mendeley.com/datasets/ppgtgvyttm/2 (doi: 10.17632/ppgtgvyttm.2).

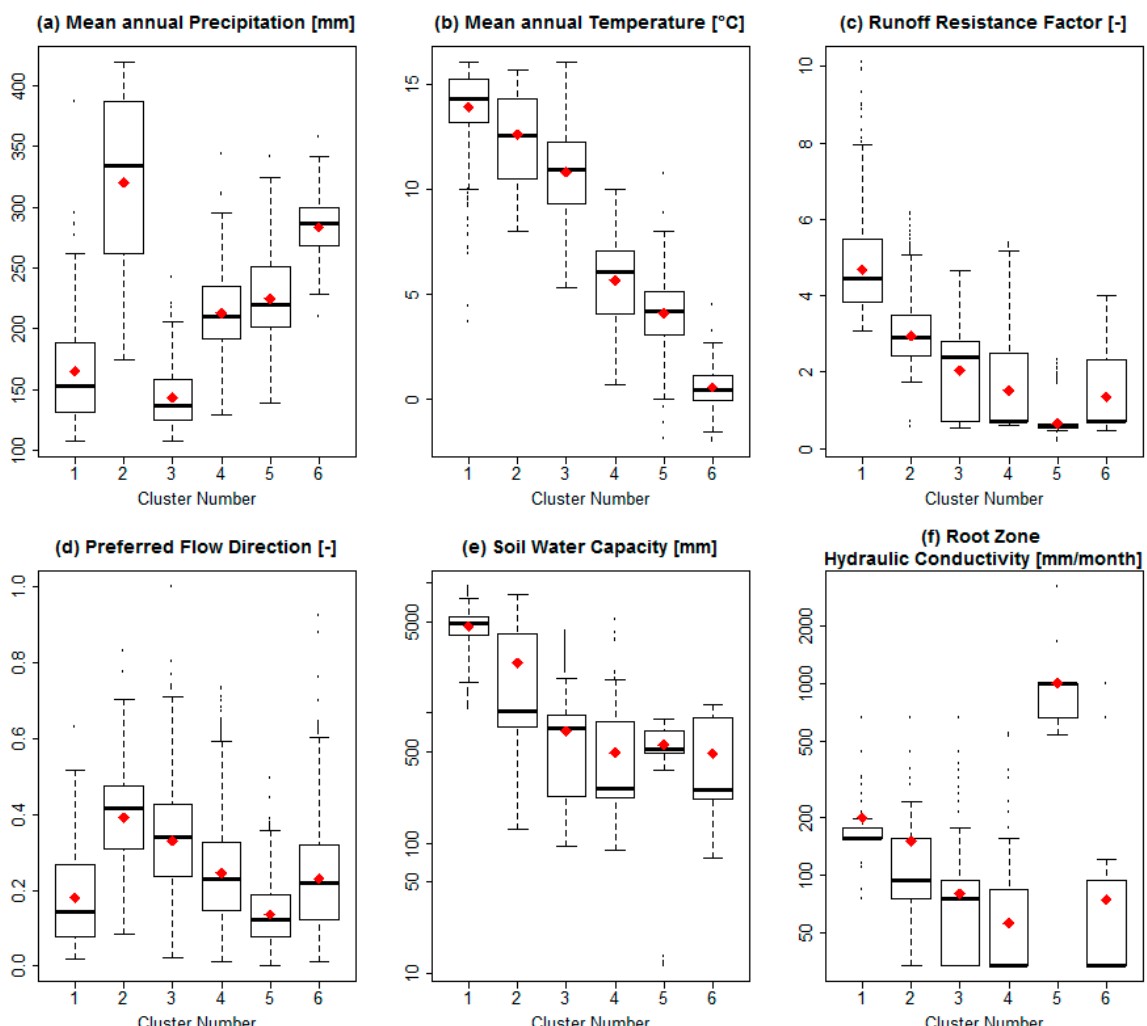

**Figure 6.** Boxplots per cluster in the simulation HRU_06 for: (**a**) Mean annual precipitation, (**b**) mean temperature, (**c**) RRF, (**d**) f, (**e**) Sw and (**f**) Ks. Red circles represent the mean value of each cluster. Sw and Ks are plotted on a log scale. Means are shown as red dots.

### 3.2. Hydrological Modeling and HRU Contribution

The WEAP model was run using ten scenarios, each one with a different number of HRUs, ranging from 1 to 10 (labeled as HRU_01 to HRU_10), as explained in Section 2.4.

Each simulation was run in a monthly time step starting from April 1979 until March 2016, following the water year commonly used in Chile, although the first four years were dismissed in the analysis due to the warming period of the model.

Table 2 shows performance scores for the results of the hydrological simulations for each run. The results are considered acceptable as they are similar to those from other WEAP simulations in the Chilean semi-arid climate, but using fewer HRUs. For example, in Vicuña et al. [15], several sub-basins of the Limarí River Basin (a basin roughly 200 km north of La Ligua River Basin) were modeled in a monthly time step with the number of HRUs between 12 and 35, depending on the land use classification and elevation range and with NSE values between 0.59 and 0.76 and biases ranging from 2.2% and −8.8%. In another work from Bonelli et al. [14], the Maipo River Basin and its sub-basins (roughly 200 km south of La Ligua River Basin) were modeled and NSE values ranged from 0.60 to 0.77, with similar methodology and numbers of HRUs to the previous example.

**Table 2.** Performance of WEAP model in the 10 scenarios using the new methodology. NSE stands for Nash–Sutcliffe efficiency coefficient and RMSE for root mean square error.

| Scenario | NSE | RMSE |
|---|---|---|
| HRU_01 | 0.58 | 4.1% |
| HRU_02 | 0.71 | 4.3% |
| HRU_03 | 0.72 | 3.6% |
| HRU_04 | 0.77 | 3.5% |
| HRU_05 | 0.78 | 3.2% |
| HRU_06 | 0.79 | 3.1% |
| HRU_07 | 0.78 | 3.1% |
| HRU_08 | 0.77 | 3.1% |
| HRU_09 | 0.76 | 3.2% |
| HRU_10 | 0.74 | 3.2% |
| Goal | 1.00 | 0.0% |

Table 2 also shows that as the number of HRUs increases and the level of spatial discretization becomes more detailed, the model efficiency also increases. For the first simulation with a lumped scheme, the NSE and RMSE (Equations (2) and (3)) values were 0.58 and 4.1%, respectively, and both indexes improved as more HRUs were used. However, for simulations with more than six HRUs, the extra clusters or HRUs are not making any considerable improvement to the results, consistent with what was described in Haverkamp et al. [11] and the model efficiency fluctuates at a plateau of 0.76–0.79, while the RMSE is near 3.1–3.2%. Figure 7 shows the observed and simulated monthly hydrograph for HRU_06.

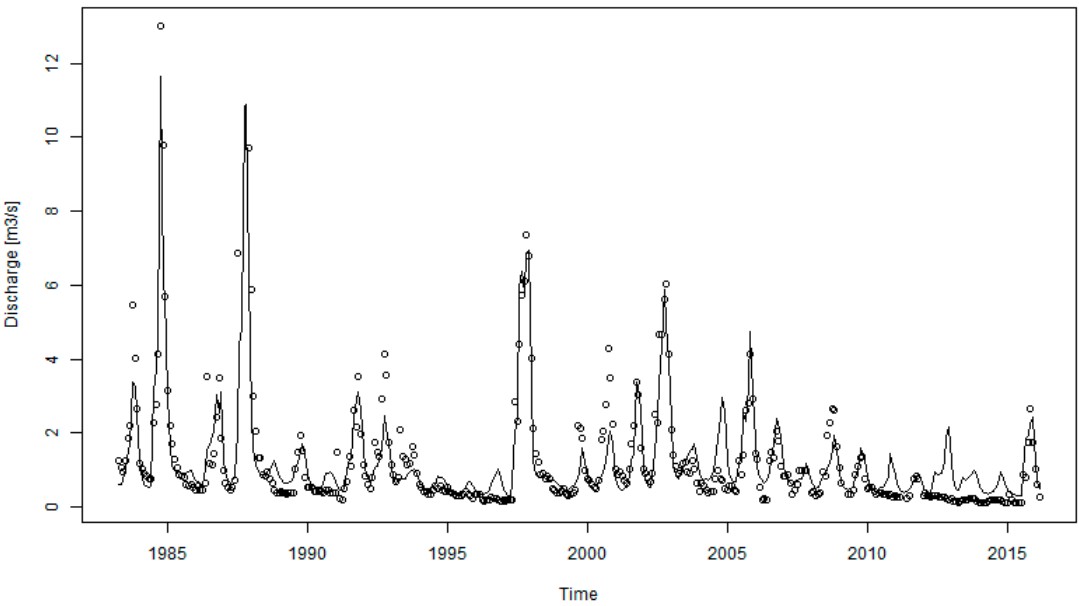

**Figure 7.** Hydrograph of observed and modeled streamflow in "Rio Alicahue en Colliguay" station for the simulation HRU_06. Observed discharge is shown as dots and the simulated discharge as a continuous line.

Table 3 presents the differences between clusters in terms of inputs and responses and it is used to assess the different hydrological processes that each HRU represents. It shows the mean annual temperature, rainfall and elevation, the mean annual discharge and its standard deviation, the variation coefficient and the centroid of the annual flow volume as an index to measure the timing of peak

discharge in the season, calculated as the weighted average of the month and the discharge associated with each month [12]:

$$HydroCentroid = \frac{\sum_{i=1}^{N} month\_index_i \cdot Q_{si}}{\sum_{i=1}^{N} Q_{si}}$$

(4)

where January has index 1 and December, 12.

**Table 3.** Summary of mean annual variables for the six clusters used in the hydrological simulation. The values correspond to the mean values for the simulation period of 1984–2016.

|  | Cluster 1 | Cluster 2 | Cluster 3 | Cluster 4 | Cluster 5 | Cluster 6 |
|---|---|---|---|---|---|---|
| Area (km$^2$) | 34.0 | 22.1 | 125.2 | 121.7 | 10.9 | 41.5 |
| % over total area | 9.6% | 6.2% | 35.2% | 34.2% | 3.1% | 11.7% |
| Elevation (m.a.s.l.) | 1483 | 1460 | 2063 | 2080 | 2837 | 2951 |
| Precipitation (mm) | 204 | 413 | 175 | 269 | 287 | 357 |
| Evapotranspiration (mm) | 126 | 203 | 110 | 154 | 50 | 161 |
| Evapotranspiration/Precipitation (–) | 0.62 | 0.49 | 0.63 | 0.57 | 0.17 | 0.45 |
| Discharge |  |  |  |  |  |  |
| Mean (m$^3$/s) | 0.09 | 0.15 | 0.26 | 0.45 | 0.08 | 0.26 |
| % over total discharge | 7% | 12% | 20% | 35% | 6% | 20% |
| Standard deviation (m3/s) | 0.03 | 0.11 | 0.30 | 0.77 | 0.08 | 0.47 |
| Coefficient of variation | 0.34 | 0.76 | 1.16 | 1.72 | 0.99 | 1.79 |
| Hydrograph centroid (month index) | 9.65 | 9.85 | 9.79 | 10.57 | 10.35 | 11.49 |

Although clusters 6 and 3 have similarities in total discharge, for its annual variation, cluster 6 groups most of the coldest cells in the basin where the snow melts late during the season and its discharge center of mass is in the middle of November and peaks in January, while cluster 3 peaks in the middle of September, coinciding with its center of mass on the hydrograph. Both clusters are controlled by very different hydrological process and parameters.

Of interest is cluster 5, as it has a relatively small amount of area but its proportional contribution to the total discharge doubles its relative area. It peaks at the beginning of summer, has the lower evaporation/precipitation ratio and it is an example of an extreme hydrologic behavior that must be characterized and not dismissed. It is possible to argue that clusters 5 and 4 can be merged, as they peak at the same time and they have the same elevation, but that decision depends on to what extent it is possible to aggregate.

Clusters 1 and 2 are the lower in elevation, but the relative contribution of cluster 1 compared to its relative area indicates that it is less important and its discharge is comparable in absolute terms to cluster 2, although their areas are 34.0 and 22.1 km$^2$.

Finally, Figure 8 presents the mean monthly discharge (left) and the mean monthly areal discharge production (right) for each of the six clusters. Each cluster shows different hydrographs in terms of total volume and peak timing, consistent with the goal of the maximization of the between variability; this behavior is also seen in other scenarios with different numbers of clusters, although not shown here. Cluster 4 is the main contributor to the annual discharge, and it is clear it has a nival hydrologic regime with its peak in November (mid-spring in the Southern Hemisphere), coinciding with the basin peak due to snowmelt. Clusters 1–3 present hydrographs with peaks in or near September, two months later than the precipitation peaks for this region during austral winter, probably due to the first snowmelts but also from interflow produced from rainfall that reacts slower than direct runoff. Cluster 1 is also the more stable in terms of discharge, mainly due to the ability to hold water because of its larger soil water capacity and cluster 3 presents a more distinct peak at the end of winter, probably due to the first snowmelt but also rainfall and humidity leaving the upper part of the soil. Cluster 6 presents the most retarded hydrograph peak in the season, explained by the late melt of snow due to its relatively higher mean elevation compared to the other HRUs, hence, the lowest values of mean temperatures. Clusters 4 and 5 also present a nival regime as their peaks match with the snow melting season, but the total volume is completely different as cluster 5 is explained by a concentration in a relatively small area of

marshes and Andes wetlands, while cluster 4 shows the biggest contribution to the total streamflow, mainly given by its large proportion of area (34.2%).

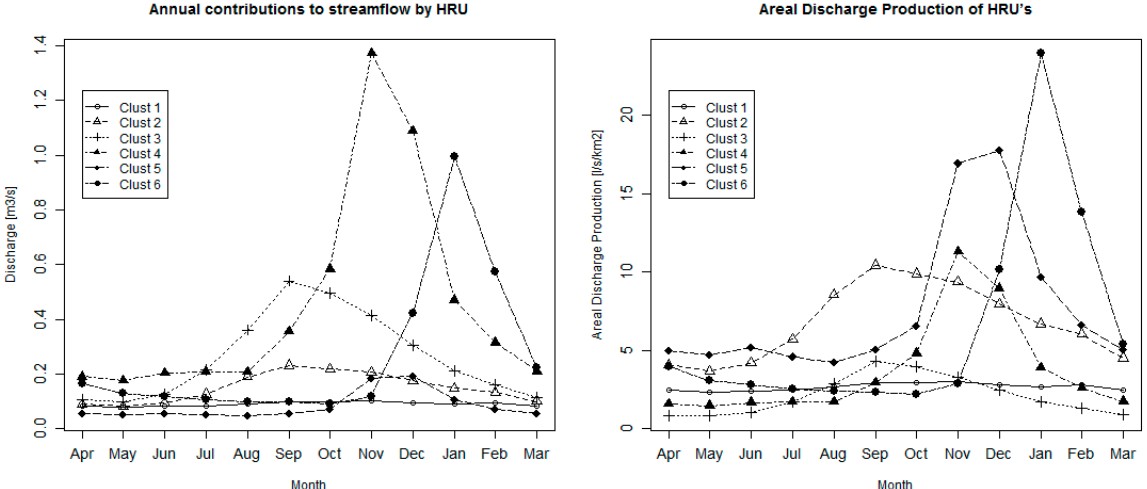

**Figure 8.** Mean monthly discharge of each HRU contributing to total streamflow at the outlet, for the six HRUs scenario. (**Left**) mean annual discharge, (**right**) mean annual discharge production by area.

The image on the right shows the production of discharge relative to the area of the HRU (in l/s/km$^2$). The relative importance of each HRU changes, especially for clusters 2, 5 and 6. As shown in Table 3, the relative contribution to the total discharge of those clusters doubles their area relative to the total area. The hydrologic regime of cluster 2 tends to be closer to the precipitation season (May to August) and it has a high areal production of water due mainly to the concentration of rainfall in that area of the catchment. Cluster 5 presents the higher average of areal production (7.6 l/s/km$^2$) and even its base value of nearly 5.0 l/s/km$^2$ is also higher than the rest of the base values. Again, this may be explained by the nature of the vegetation covering most of that cluster and by the slow release of water stored in it. The highest peak of 23.9 l/s/km$^2$ in the month of January corresponds to cluster 6 and it is a combination of high rates of precipitation during winter in a relatively small area of the catchment, accumulating a massive volume of snow with the rise of temperatures in summer, producing the highest peak of discharge per unit of area due to snowmelt.

## 4. Discussion

This paper presents a new methodology for HRU delineation based on the catchment attributes, explained by the model parameters and climate variables. The units generated are expected to be used in lumped and semi-distributed hydrological models where the topology of the elements could be neglected. The methodology present two main steps: (i) A principal component analysis to reduce the number of variables while most variance is kept and (ii) a hierarchical clustering decomposition to delineate the HRUs with minimum internal variability but maximum variability among the created units.

The methodology was tested on the Alicahue River Basin, a small basin located in a semi-arid region in Central Chile, using the WEAP model, which has a hydrology module. The model was run under ten scenarios with different numbers of clusters (HRUs) and evaluated using the Nash–Sutcliffe efficiency index and the mean square error.

### 4.1. Methodology and Data Uncertainties in the Dataset Preparation

Although the generation of the complete dataset used to derive the HRUs was not presented, the estimation of the parameters and the climate dataset can be calculated independently using any methodology or information previously made available. The only two main characteristics that must

be preserved from such a methodology are: (1) Climate information needs to be from a gridded dataset and (2) hydrologic parameters must be specific to the target model and calculated spatially prior to the HRU delineation.

The main reason to use a WRF simulation, instead of longer and publicly available datasets, was its high resolution (1 km) which is very important in regions with very complex topography as the Alicahue basin. That resolution is much higher than Reanalysis [30] which is usually about 0.5° and even higher than the newest local dataset CR2MET available at 0.05° for the continental Chilean territory [31,32].

However, it is important to highlight the evident limitation in the case study due to having only 3 years of WRF simulations to represent the climate of the region. For example, the analysis of the Hurst parameter of the climatic dataset could be useful to improve the climatic characterization of the studied area, as the Hurst–Kolmogorov (HK) behavior is characterized by strong spatio-temporal correlation in a vast range of scales, as shown by global analyses of hydrometeorological processes [33–35]. Dimitriadis and Koutsoyiannis [36] also illustrate how the HK behavior highly increases the variability in scale, an attribute that can affect the calculation and selection of the HRU.

## 4.2. Clustering Method and Results

Once the dataset was pre-processed, it contained more than 17 thousand cells and each cell had information on four hydrologic parameters and seven meteorological variables; moreover, in other implementations, this number could be even bigger. That amount of data had to be summarized in order to be treatable, but at the same time, it was desirable that the aggregation process carried most of the variability, without losing valuable information. The principal component analysis was chosen, as it selects orthogonal vectors that carry much of the information gathered in the previous process.

The PCA function uses weights to account for the relative importance of the variables. This gives the modeler the option to assess the most important variables given the model and/or the problem to solve. In this study case, weight for rainfall and temperature was equal to 2, while for all other variables, it was set to 1. Temperature plays a crucial role in controlling evapotranspiration and snow melting, both main hydrological characteristics of an Andean semi-arid basin, such as the Alicahue Basin. Precipitation controls the water coming into the basins and water simulations are highly sensitive to the amount of water used in the model. Those weights give a chance to the modeler to exert major control over the importance of each variable/parameter in the delineation of the HRUs and even to perform a sensibility analysis of those weights.

The within variance decrease obtained from the clustering process could be used as a criterion for the selection of the optimal number of HRUs required to capture the main hydrological behaviors in the target basin. This methodology allows for a reduction in the number of HRUs involved in the simulation, decreasing the required computational time. This could favor, for instance, studies with ensemble simulations, more exhaustive sensitivity analysis for some parameters of the models and/or much longer (or higher temporal resolutions) simulations.

## 4.3. Discharge Independence in the Hydrologic Modeling

The results by the hydrological modeling show, in general, a good match between observations and simulations, even with the lumped scheme. As expected, the simulation with one HRU, as the most lumped scheme, shows the poorest results in terms of efficiency (NS = 0.58). As the number of HRUs increases and the level of spatial discretization is more detailed, the model efficiency also increases. The errors in the modeling can be explained, at least to a large extent, by the uncertainties in the simple meteorological models used to derive the precipitation and the temperature, the proposed relations to obtain the parameter maps and a possible lack of representation of all the hydrologic fluxes and storages in the hydrologic cycle in the basin due to a lack of soil information. Additionally, the representation of extreme hydrologic phenomena is possible only if the chosen model is capable of

simulating these phenomena. If not, any discretization of the HRU methodology would be useless or at least less useful.

The streamflow at the outlet of the Alicahue Basin is controlled by a baseflow dominated by the subsurface storage which is dependent on the storage capacity of the soil and evapotranspiration stress, a component driven by the winter rainfall dependent on hydraulic conductivity and rain intensity, and a component driven by snow melting which is highly dependent on temperature and elevation. From Figure 8, such behaviors were well captured for the different HRUs, which gives the modeler a better understanding of the underlying processes controlling the outlet streamflow when compared to other methodologies for HRU delimitation.

Finally, it is important to note that the only variable for assessing the goodness of fit of the hydrological model was the river discharge, which simplified the analysis of the water cycle. It would be advisable to test the methodology and the hydrologic behavior of the rest of the components of the hydrological cycle (infiltration, evapotranspiration, groundwater movement, leakage, etc.), which was not possible in the case of the study basin due to the lack of observations.

## 5. Conclusions and Further Developments

Flügel [2] presented in 1983 the concept of HRUs for hydrological modeling. HRUs are the basic units in which the equations controlled by parameters are run and meteorological data are used as inputs. The basic assumption of HRUs is that each of them has a particular hydrological response to rainfall, temperature and other climate data. Most of the current methodologies account only partially for the spatial variability that leads to a differentiated response, and particularly the spatial climate variability within the basin is under- or misrepresented.

This paper presented a methodology for the determination of HRUs, more consistent with the classical definition, based on hydrological parameters (specific to the target model) and meteorological inputs. The methodology uses principal component analysis and hierarchical clustering to minimize the global internal variability in each HRU and at the same time to maximize the variability among HRUs. This procedure is intended to generate different responses in each unit, defined by the modeled hydrograph, minimizing the number of required HRUs to capture the hydrologic behavior of the system.

The application of the methodology was assessed in the Alicahue River Basin, a small basin located in a semi-arid and mountainous region in Central Chile, with altitudes ranging from 780 to almost 4000 m above sea level. The results of the WEAP simulations show a good agreement between modeled and observed streamflow at the outlet, with scores comparable to other studies using the same model in similar basins.

The main advantages of the proposed methodology can be summarized in three concepts: Improvement in computational efficiency, basin heterogeneity captured and optimization and identification of the HRUs. As the methodology is designed to minimize the required numbers of HRUs to account for most of the spatial variability in the climate and hydrological parameters (the main controllers of the hydrological response), the computational effort is highly reduced, as computational time is usually negatively correlated to the number of HRUs. In the study case, only six HRUs were necessary to achieve similar scores to those from more common methodologies that use several tens of HRUs. For the second concept, the PCA captures most of the variability of the parameters and climate variables (some of them being spatially and temporally correlated) and heterogeneous conditions of the data are kept even after the reduction of the number of variables used as input to the cluster analysis. Additionally, the hierarchical clustering process ensures that the delineation of the HRUs is completely driven by the information in the eigenvectors and not by arbitrary choices and is not influenced by noise from the original variables. For instance, in the study case, one of the HRUs corresponds to a small and disjointed area that has a relatively large contribution to the total streamflow, which would probably be neglected with most of the traditional methodologies. Lastly, as the HRU delineation was driven by the minimization of the within variance in each HRU and,

at the same time, maximization of the variance between HRUs, the hydrological response is expected to be different for each HRU with minimum redundancy. This allows the modeler to gain a better understanding of the underlying hydrological behaviors that control the response of the basin: each HRU in the case study was identified with different processes, including baseflow, quick rainfall–runoff response, snow melting at different times associated with elevation and temperature differences.

Better hydrological parameters and meteorological datasets could still improve the model efficiency and hydrologic understanding of the basins. For example, it would be advisable to consider long-term climatic records with high spatial resolution to study the Hurst–Kolmogorov behavior of the meteorological and hydrological variables, as the distribution and localization of the calculated HRUs may be more robust and stable. A deeper comparison between spatial resolution (i.e., smaller cell grid), temporal availability and robustness of the data must be assessed, especially for small and mountainous river basins as in this case study.

Another future research direction is to test the methodology in basins with different hydrologic regimes and using different models. WEAP is suitable for time steps longer than one day, but the methodology could be used in other long-term models or even in storm models, considering other parameters sensible to the basin response (i.e., concentration time, curve number, etc.).

**Supplementary Materials:** Supplementary data is available at: https://data.mendeley.com/datasets/ppgtgvyttm/2 (doi: 10.17632/ppgtgvyttm.2).

**Author Contributions:** D.P. and J.A.: Conceptualization, Data curation, Formal analysis, Investigation, Methodology, Software, Validation, Visualization, Writing—original draft and Writing—review and editing. O.N.: Formal analysis, Investigation, Software, Visualization and Writing—original draft. F.F.: Data curation, Visualization and Writing—original draft. All authors have read and agreed to the published version of the manuscript.

**Funding:** This research received no external funding.

**Conflicts of Interest:** The authors declare no conflict of interest.

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
