# Peer review of "Optimization of Hydrologic Response Units (HRUs) Using Gridded Meteorological Data and Spatially Varying Parameters"

_water, doi:10.3390/w12123558_

Round 1
Reviewer 1 Report
The paper presents new methodology names PCA/HCPC Method to divided catchment to optimal number of Hydrological Response Unit HRU. The new methodology was used to delineate HRU in WEAP model and was testing in The Alicahue river basin, Chile. The paper presents significantly novelty in modelling of hydrological phenomena by used PCA/HCPC Method to divide catchment on optimal number of HRU. Paper is interesting, well written and organizing. I have some minor suggestion to improve this work:
1 L 150 should be m3/s
2 L166-175: I suggest adding scheme of WEAP model. On this fig can be showed interactions between each components in model
3 Results section: I suggest showing quality of model based on criteria described by Ritter, A., Muñoz-Carpena, R. (2013). Performance evaluation of hydrological models: statistical significance for reducing subjectivity in goodness-of-fit assessments. J. Hydrol., 480, 33–45
4 L360-366: Please show fig or table where will be all results of quality of model for each variants of HRU. This can help to show evolution of testing model
5 Conclusion L516-518: But in this work I can not see comparing of simulations with proposed methodology with other studies. Please show this comparison in discussion section or in results where can be compare different methodologies of HRU identification
Reviewer 2 Report
This work presents a methodology for defining the smallest appropriate number of hydrological homogenized clusters, in terms of the variability of the hydrometeorological variables in a river basin, by minimizing the internal variance of each cluster and by maximizing the total variance among clusters. The analysis is interesting and important in hydrological studies for a river basin. Please see some major and minor comments and suggestions below that may be helpful for the revision of the paper:
1) Please give more information about the WRF simulations (including that the initials stand for Weather Reaserach and Forecasting). Since the applications are based on these simulations, it would be useful for the Readers to learn more about this.
2) Although the lack of long-term records is mentioned in the text, it may be appropriate to add a paragraph in the text concerning any implications to the current study. In case the analyzed hydrometeorological variables (streamflow, precipitation, temperature, relative humidity, wind velocity, albedo and evapotranspiration) exhibit a white-noise or a Markov-type behaviour, then there would be no effect from the short length of the analyzed spatio-temporal timeseris. However, as empirically shown from global analyses of thousands of stations and/or from massive long records timeseries, the aforementined processes exhibit a long-term persistence behaviour or else called Hurst-Kolmogorov behaviour (see an extended review, global analyses of thousand of stations, and/or analysis of massively long timeseries of several decades, for temperature, wind speed, precipitation, streamflow, solar radiation -which is directly linked to the albedo-, evapotranspiration -which is mostly affected by the temperature and wind-, and dew-point -which is directly linked to relative humidity- in DOI:10.13140/RG.2.2.34652.69768; also for the physical justification through entropy maximization see doi:10.1007/978-3-319-58895-7_14). The HK behaviour may highly increase the variability of the process in scale (for example, see Fig. 2 in doi:10.1007/s00477-018-1540-2, where the variability of the estimated first four central moments are plotted vs. the length of the sample timseries), and therefore, it could be discussed in the text that for the selections of the HRU clusters to be stable long-term records may be required for the hydrometeorological variables.
3) For the PCA, it is metioned in the text that it requires for the variables to be uncorrelated. However, all the variables mentioned in the previous comment exhibit a strong spatio-temporal auto-correlation as well as cross-correlation. Please consider giving more information of what exactly are the implications in the analysis from the assumption of independency.
4) Also for the PCA analysis, it is mentioned in the text that a weight of 2 is assigned to temperature and precipitation. Please consider further explaning why a factor of 2 (and not for example 1.5 or 3) is assigned to these two variables, or to perform a sensitivity analysis on the weights in order to justify this decistion.
5) In the text, it is mentioned that each simulation ranged from April 1979 to March 2016. What climatic conditions did you use for each simulation, since only a couple of years of records were available for the meteorological variables?
6) In section 3.2 for the WEAP model, it is mentioned that: "But for simulations with more than six HRUs, the extra clusters or HRU are not making any considerable improvement in the results". Please consider showing results from more than six simulations, so that it can be clear to the Readers why you used up to six HRUs.
7) In the text, it is mentioned that the only variable for the methodology assessment is the streamflow, which simplify the water cycle and all its components into one lumped criterion. I am not sure I understand this statement, since more hydrometeorological variables were analyzed and not just streamflow. The observed variability in the analysis is due to all the above variables and not just the discharge. Please consider further discussing this in the Discussion.
Round 2
Reviewer 2 Report
The Authors have addressed all the comments and suggestions. Please consider providing some additional information concerning the 2nd and 3rd comment of the previous review:
2) The Hurst-Kolmogorov (HK) behaviour is characterized by strong spatio-temporal correlation in a vast range of scales, as shown by global analyses of hydrometeorological processes such as the ones used in the analysis. Consider mentioning a work from the literature that describes in more detail this behaviour. The Authors may mention any such reference. For example, Koutsoyiannis et al., (2018) contain both a mathematical description and justification as well as global analyses about the HK behaviour, whereas Dimitriadis and Koutsoyiannis (2018) illustrate how the HK behaviour highly increases the variability in scale; an attribute that will certainly affect the selections of the HRU clusters that is currently based in the assumption that "...the variation of the hydrological process dynamics within a single HRU is small compared with the hydrologic dynamics and responses to other units defined in the model.". In other words, since the hydrometeorological variables used here are characterized by the HK behaviour, then the locations of the selected HRUs may alter if longer timeseries were available to the Authors for their analysis.
Dimitriadis, P., and D. Koutsoyiannis, Stochastic synthesis approximating any process dependence and distribution, Stochastic Environmental Research & Risk Assessment, 32 (6), 1493–1515, doi:10.1007/s00477-018-1540-2, 2018.
Koutsoyiannis, D., P. Dimitriadis, F. Lombardo, and S. Stevens, From fractals to stochastics: Seeking theoretical consistency in analysis of geophysical data, Advances in Nonlinear Geosciences, edited by A.A. Tsonis, 237–278, doi:10.1007/978-3-319-58895-7_14, Springer, 2018.
3) In the reply to the 3rd comment, the Authors mention that the PCA method generated uncorrelated new variables. It is still not clear to me why did you generated uncorrelated variables with the PCA since (a) the PCA can also generate correlated variables as the Authors mention, and since (b) the hydrometeorological variables used in the analysis may exhibit strong spatio-temporal correlation. I only persist on this because it could be difficult for the Readers to understand.
